# The Role of Hypertonic Saline in Ablative Radiofrequency of the Sacroiliac Joint: Observational Study of 40 Patients

Ezio Amorizzo [1,2] and Gianni Colini-Baldeschi [1,*]

1   Pain Clinic Roma, 00135 Rome, Italy
2   S. Paolo Hospital, 00053 Civitavecchia, Italy
*   Correspondence: g.colini@libero.it

**Abstract:** Background: The aim of this retrospective uncontrolled article is to illustrate a technique of neurotomy of the sensitive branches of S1 S2 S3 in RFA that appears to result in a better success rate and longer-lasting pain relief. Methods: 40 patients were treated, 26 females and 14 males, with an average age of 74 (92–55). After the examination, the patients underwent an ultrasound-guided diagnostic block of the affected sacroiliac joint. Only patients who presented pain relief greater than 60% after the diagnostic block were candidates for the RFA procedure. The procedure was always performed in the operating room on an outpatient basis. After obtaining the best fluoroscopic visualization of the joint to be treated, two RFA cannulae were placed starting from the lower medial margin parallel to the SIJ to perform a bipolar RFA along the entire medial margin of the SIJ. Lidocaine 2% and hypertonic saline 2 mEq/mL were used for each RFA level. Patients were followed-up at 3, 6, 12, 18, and 24 months by evaluating the NRS and SF-12. Results: Patients reported extreme satisfaction with the procedure performed and reported a significant improvement in NRS and SF-12 at FU visits. No adverse events occurred. Conclusions: Bipolar RFA treatment of the sacroiliac joint with the use of a hypertonic saline solution appears to improve the success of the method and its durability. We are inclined to believe that the use of hypertonic saline may significantly increase the lesion area and result in a greater effect on the sensory branches.

**Keywords:** radiofrequency ablation; chronic pain; hypertonic saline; nociceptive pain

## 1. Introduction

Sacroiliac joint dysfunction (SIJ) is a very common cause of refractory lower back pain, although often underestimated and undiagnosed. Sacroiliac joint pain is estimated to account for 20 per cent of all causes of lower back pain. SIJ pain is commonly felt in the gluteal region but may also affect the lower extremities and/or the lumbar region. The sacroiliac joint, a diarthrodial joint, connects the sacrum and ilium. It is a very stable joint due to the presence of anterior, interosseous, and posterior ligaments. The posterior sensory innervation of the SIJ comes from the dorsal branches of L5 S1 S2 S3 with components that can also come from L4 and S4. During the physical examination, patients with sacroiliac joint problems show at least one (or all) of the following symptoms: lower back pain (below L5), pelvic/gluteus pain, hip/inguinal pain, discomfort in a sitting position (inability to sit for long periods of time or sitting on one side only), or pain when moving from a sitting to a standing position. Differential diagnosis to exclude other causes of pain: an absence of pathologies referable to the lumbar spine and hip. In an upright position, the patient should indicate the point where they feel pain, and experience pain at the groove of the sacroiliac joint and on the weight-bearing side following the one-legged balance test. Patients may have pain upon palpation of the sacroiliac joint, present positivity on provocation tests for at least three out of five tests (distraction, thigh thrust, FABER, compression, Gaenslen's test), and have a significant reduction of pain after SIJ diagnostic block [1–3]. Conventional treatment for chronic SIJ pain includes medication, physical therapy, and, in the case of

a positive response to a diagnostic block, radiofrequency ablation (RFA) of the sensory branches. Numerous techniques have been proposed to perform an RFA with a success rate, reported in the literature, of 50% for an average duration of 6 months after the procedure [4–6]. The aim of this article is to illustrate a technique of neurotomy of the sensitive branches of S1 S2 S3 in RFAs that appears to result in a better success rate and longer-lasting pain relief.

## 2. Methods

This 40-patient retrospective observational cohort study considered RFA treatments of the SIJ performed at 2 pain therapy units. All procedures were performed by the authors, taking into consideration the cases treated from 2019 to 2021. In total, 40 patients were treated, 26 females and 14 males, with an average age of 74 (92–55). All patients experienced sacroiliac pain, confirmed by clinical examination, with an average NRS of 8.2. Patients had lower back pain that lasted longer than 3 months, presenting positive provocation tests for at least 3 out of 5 tests, and were not responding to conventional medical treatments. Autoimmune pathologies were excluded by performing serological tests. No other possible causes for back pain have been identified. Exclusion criteria were coagulopathies, systemic infections or infections at the site of the procedure, pregnancy, ages of less than 18 years, severe psychological disorders, allergies to local anesthetics, and senile dementia. After the examination, the patients underwent an ultrasound-guided diagnostic block of the affected sacroiliac joint with 2 ccs of 2% lidocaine. Only patients who presented pain relief greater than 60% after the diagnostic block were candidates for the RFA procedure. Informed consent forms were collected for both the procedure of the diagnostic block and the neurotomy in the RFA. The RFA procedure was always performed using a Boston Scientific G4 Radiofrequency Generator, in the operating room on an outpatient basis, and under constant fluoroscopic vision. The skin was prepped and draped. The patients were treated in the prone position and landmarks were identified under fluoroscopic guidance. In most of the cases, the RFA was performed under conscious sedation or without any sedation. After obtaining the best fluoroscopic visualization of the joint to be treated, 2 RFA all-in-one electrode, cannula and injection tubes, 100 mm 20 G AT 10 mm, were placed starting from the lower margin parallel to the SIJ with a distance between the needles of 0.81 cm to perform a bipolar RFA along the entire medial margin of the SIJ (average of 5 treatment levels) moving the distal cannula cranially for each level (Figure 1). A motor and sensory stimulation test was always performed prior to the lesion at 50 Hz and 2 Hz, respectively, with a sensory perception threshold within 0.5 V. The first 2 cannulae were then injected with 1 cc lidocaine 2% and 1 cc hypertonic saline 2 mEq/mL. In the subsequent lesions, lidocaine and hypertonic saline were injected into the cephalically repositioned cannula only. The lesion was performed at 72 °C for 90″ for each pair of cannulae at all treated levels. At the end of the procedure, the patients were discharged after about 30–45 min.

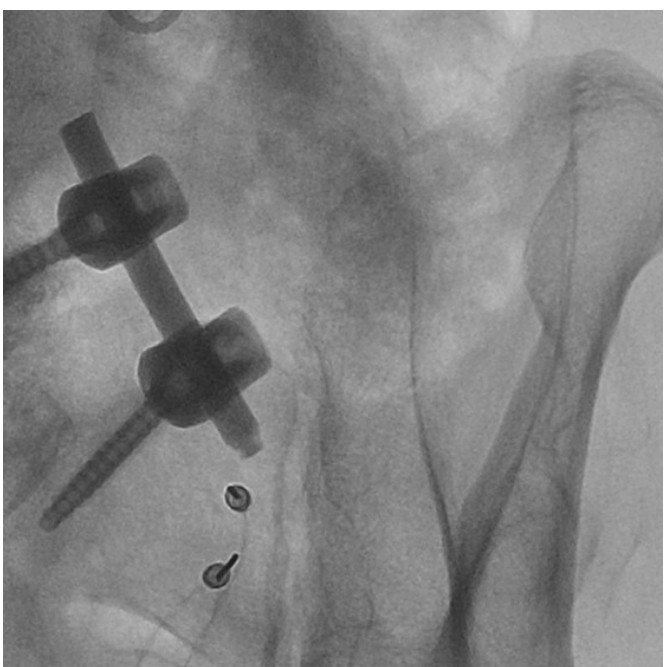

**Figure 1.** An example of a level of cannulae placement.

## 3. Results

All patients were followed-up with FUs at 3, 6, 12, 18, and 24 months by evaluating the NRS and SF-12. In total, 4 patients completed the FU at 3 and 6 months. All 40 patients who underwent RFAs reported excellent pain relief at 3- and 6-months FUs (over 80% reduction in NRS) and stopped taking analgesic medications. The mean NRS value at 3 months was 1.3, at 6 months 1.5. A total of 36 (90%) patients completed FUs at 12, 18 and 24 months. At the 12-month FU, 4 (11%) of 36 patients reported good pain relief (40% to 50% reduction in NRS), while the remaining 32 (80%) patients reported excellent pain relief (over 80% reduction in NRS). The mean NRS value at 12 months was 2.3. At the 18-month FU, 10 (27.7%) of 36 patients reported good pain relief (40% to 50% reduction in NRS), while the remaining 26 (72.2%) patients reported constant excellent pain relief (over 80% reduction in NRS). The mean NRS value at 18 months was 3.2. At the 24-month FU, 17 (47.2%) of 36 patients reported good pain relief (40% to 50% reduction in NRS), while the remaining 19 (52.7%) patients reported excellent pain relief (over 80% reduction in NRS). The mean NRS value at 24 months was 3.4 (Figure 2). Patients who reported good pain relief at the 12–18- and 24-months FUs significantly reduced their analgesic medication intake by discontinuing continuous intake. All patients who reported excellent pain relief discontinued their previous pain medication completely. The mean value of the SF-12 administered before treatment was PCS 28.5, and MCS 36.9, respectively, while the mean value in the FU up to 24 months was PCS 46.7 and MCS 58.5, with an improvement in PCS of 163.9% and MCS of 158.5% (Figure 3). We also compared the NRS values of the patients before treatment and at 24 months which showed a 1-tailed t-value of $-18.71$ with statistical significance with a *p*-value $< 0.05$. Patients reported extreme satisfaction with the procedure performed and reported a significant improvement in their quality of life. Furthermore, they reported that they would repeat the procedure in case of recurrence of pain symptoms. No adverse events occurred.

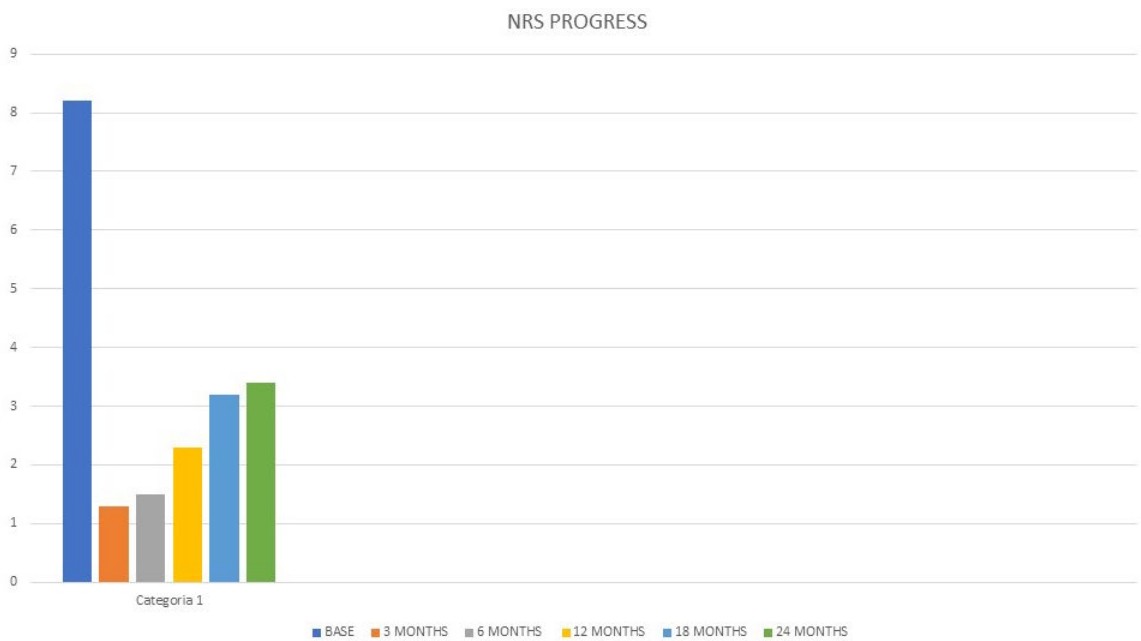

**Figure 2.** NRS Progress. NRS: Numerical Rating Scale.

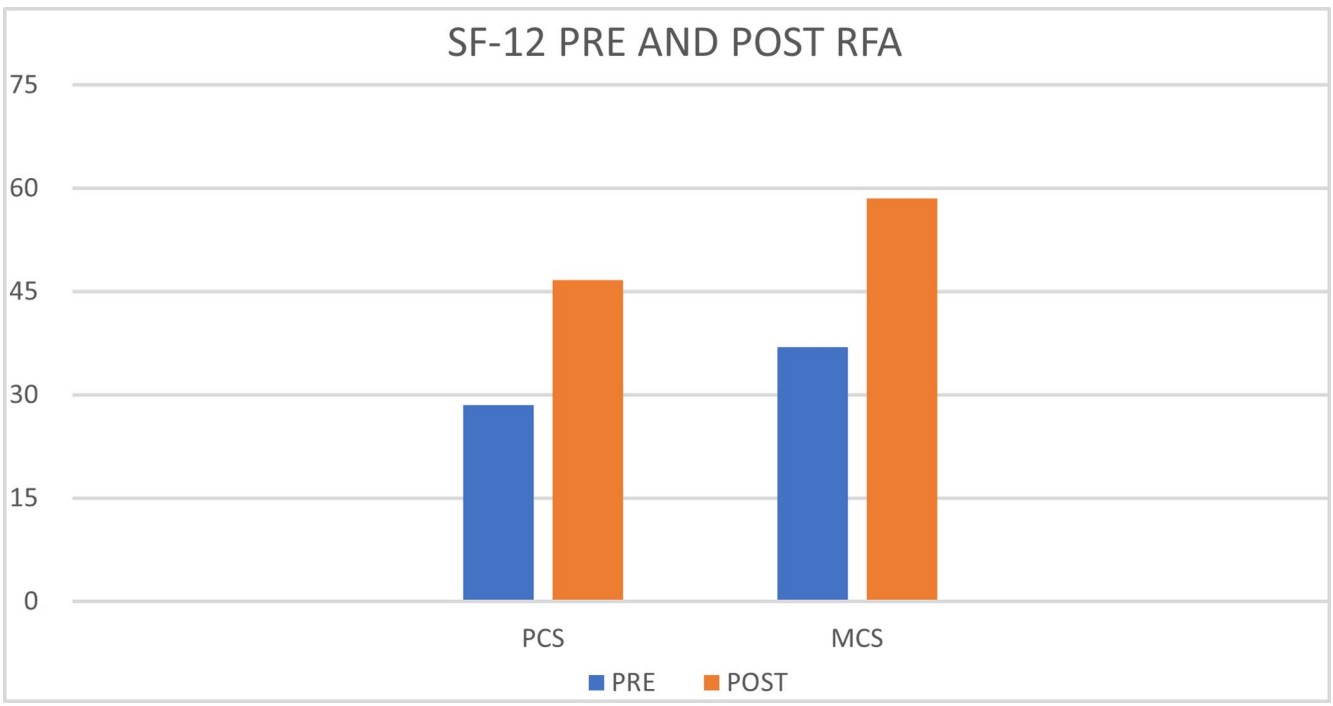

**Figure 3.** SF-12 Pre and Post RFA. SF-12 questionnaire, Physical Component Score (PCS) and Mental Component Score (MCS).

## 4. Discussion

Chronic pain syndromes of the sacroiliac joint are very often underestimated, and the data in the literature regarding pain relief after RFA treatments, performed with different techniques, generally report a 50% [7,8] improvement at 6 months. A recent review from 2021 reported variable success rates ranging from 32% to 89% in patients with SIJ pain treated with RFAs [9]. The complexity and anatomical variability in the course of the sensory branches that innervate the SIJ are certainly undeniable, and this is perhaps the reason for such conflicting results in RFA neurotomy procedures of this joint. It is certainly undeniable that the reported results are linked to the various RFA techniques used, and

these could influence the clinical outcome of the treated patients. It has been shown that pre-injection of NaCl at a concentration greater than 3% results in a larger lesion area and in bipolar RFAs results in an optimal lesion [9,10]. Taking into consideration the above mentioned, the extreme anatomical variability of the sensory branches that innervate the SIJ certainly affects the results of the techniques used and the duration of the pain relief. The bipolar technique used by performing an RFA along the entire medial margin of the SIJ, and using in addition the hypertonic saline solution, has probably allowed an increase in the area of the lesion involving most of the afferent sensory branches despite the complex anatomy. This results in a better outcome than is generally reported in the literature. Being a retrospective observational cohort study and non-controlled could be a limitation of the study itself, as well as the evaluation in the FU visits that was conducted in the pain units that performed the treatments. The reported results were evaluated based on the NRS assessment and the SF-12 questionnaire which showed, however, a higher success rate and longer duration of pain relief compared with the results reported from other studies [4–9].

## 5. Conclusions

Bipolar RFA treatment of the sacroiliac joint with the use of hypertonic saline solution appears to improve the success of the method and its durability. This is probably due to the increased lesion area with a greater possibility of involving most of the sensory components that innervate the SIJ. This modality of an RFA carried out along the entire medial margin of the sacroiliac joint has determined in treated patients a more lasting pain relief than reported in the literature, has an absence of side effects, and can also be performed on elderly and frail patients The limitation of this study is that it is a retrospective observational cohort study. These data will need to be confirmed by further studies.

**Author Contributions:** Conceptualization, E.A. and G.C.-B.; methodology, E.A. and G.C.-B.; software, E.A. and G.C.-B.; validation, E.A. and G.C.-B.; formal analysis, E.A. and G.C.-B.; investigation E.A. and G.C.-B.; resources, E.A. and G.C.-B.; data curation, E.A. and G.C.-B.; writing—original draft preparation, E.A. and G.C.-B.; writing—review and editing, E.A. and G.C.-B.; visualization, E.A. and G.C.-B.; supervision, E.A. and G.C.-B.; project administration, E.A. and G.C.-B.; funding acquisition, no funding. All authors have read and agreed to the published version of the manuscript.

**Funding:** This research received no external funding.

**Institutional Review Board Statement:** Not applicable.

**Informed Consent Statement:** Written informed consent has been obtained.

**Data Availability Statement:** The only data available are published in the article.

**Conflicts of Interest:** The authors declare no conflict of interest.

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
