# Peer review of "The Role of Hypertonic Saline in Ablative Radiofrequency of the Sacroiliac Joint: Observational Study of 40 Patients"

_clinpract, doi:10.3390/clinpract13010006_

Round 1

Reviewer 1 Report

Dear Authors,

Thank you for the opportunity to review this manuscript.  -

In my opinion, the author guidelines were not properly followed. I could not find the abstract.

-The manuscript is very brief and missing important information that should be reported.

-The methods are not adequately described. The study design is not properly descriptive.

-Statistical section is not properly described.   

-Results are not sufficiently reported.   

- The limitations of the current study are not adequately reported.  

-The overall presentation of the study is poor.  

The manuscript should be significantly improved.

Author Response

Dear Reviewer

Thank you for your time and valuable suggestions. We try to respond to your comments. We sincerely hope that what we have done meet your requirements. We would like to thank you for your time and valuable suggestions.

1)In my opinion, the author guidelines were not properly followed. I could not find the abstract.

      We followed guidelines in treating patients after excluding other possible causes of sacroiliac chronic pain or autoimmune disorders and confirmed by clinical examination. Furthermore, only those who had pain relief greater than 60% after an ultrasound-guided diagnostic block were subjected to the procedure. We have attached the abstract

2) The manuscript is very brief and missing important information that should be reported.

      Please excuse us, but since it is not specifically reported we do not know what important information   are missing

3) The methods are not adequately described. The study design is not properly descriptive.

     The method seems to us adequately described. This is just an uncontrolled retrospective study

4) Statistical section is not properly described.  

      We also compared the NRS values of the patients before treatment and at 24 months which showed a one tailed t-value of -18.71 with statistical significance with p-value < 0.05.

5) Results are not sufficiently reported.  

     Patients were followed-up with FU at 3, 6, 12, 18, and 24 months by evaluating the NRS and SF-12. Four patients completed FU at 3 and 6 months, and. All 40 patients who underwent RFA, reported excellent pain relief at 3- and 6-months FU (over 80% reduction in NRS), and stopped taking analgesic medications. The mean NRS value at 3 months was 1.3, at 6 months 1.5. 36 (90%) patients completed FU at  12, 18 and 24 months. At 12-month FU, four (11%) of thirty-six patients reported good pain relief ( 40% to 50% reduction in NRS), the remaining 32 (80%) patients reported excellent pain relief (over 80% reduction in NRS). The mean NRS value  at 12 months was 2.3. At 18-month FU, ten (27,7%) of thirty-six patients reported good pain relief ( 40% to 50% reduction in NRS), the remaining 26 (72,2%) patients reported constant excellent pain relief (over 80% reduction in NRS). The mean NRS value  at 18 months was 3.2.  At 24-month FU, seventeen (47,2%) of thirty-six patients reported good pain relief ( 40% to 50% reduction in NRS), the remaining 19 (52,7%) patients reported excellent pain relief (over 80% reduction in NRS). The mean NRS value  at 24 months was 3.4 (Fig.2). Patients who reported good pain relief at 12-18- and 24-months FU significantly reduced their analgesic medication intake by discontinuing continuous intake. All patients who reported excellent pain relief discontinued their previous pain medication completely. The mean value of the SF-12 administered before treatment was PCS 28.5, and MCS 36.9 respectively, the mean value in the FU up to 24 months was PCS 46.7 and MCS 58.5, with an improvement in PCS of 163.9% and MCS of 158.5%

6) The limitations of the current study are not adequately reported. 

     The limitation of this study is that it is retrospective and uncontrolled. These data will need to be confirmed by further studies

7) The overall presentation of the study is poor

      We hope that our corrections have made the article more complete

Reviewer 2 Report

Thank you for choosing our journal 

The article is very short and results are not clear inside the manuscript and blended with methods

Also 2 figures and no tables ( even about demographic data and comorbidities of the treated patients and it's influence on response to intervention should be clarified

Also results is very short need more expansion and discussion is very short 

Please refrase and expand discussion

Author Response

Dear reviewers
  We would like to thank you for your comments and we have tried to improve the sections you have reported. We hope that our efforts meet your expectations. Thanks again for your time and valuable suggestions

Round 2

Reviewer 1 Report

Thank you for inviting me again to review this manuscript.
The authors have significantly improved the quality of this paper but it still has many limitations (the quality of English grammar, writing and etc).

Reviewer 2 Report

Accept